# ONE DIRECTION TO RULE THEM ALL: TOWARD GENERALIZABLE SOLVING STRATEGIES ACROSS COMBINATORIAL OPTIMIZATION PROBLEMS

## ABSTRACT

Many combinatorial optimization problems (COPs) share latent structure despite differing in surface form, allowing classical heuristics to transfer with minimal adaptation. In contrast, most learning-based solvers are trained in isolation and fail to leverage cross-problem commonalities. This paper explores the possibility of learning generalized solving strategies that capture shared structures across different COPs, enabling easier adaptation to new tasks. We leverage a header-encoder-decoder architecture in which light problem-specific headers and decoders handle inputs and outputs, while a shared, heavy encoder is trained to capture problem-agnostic solving strategies. The key is to align the encoders optimization behavior across tasks by enforcing gradient consistency, making updates induced by different COP objectives point in similar directions with comparable magnitudes. We realize this via task-specific feature rotation matrices and loss weights that steer the encoders gradients, learned alongside the solver in a bi-level procedure: an inner loop optimizes each task with reinforcement learning on its true objective, and an outer loop tunes rotations and weights through a gradient consistency loss. Experiments on six COPs show that it enhances the model's ability to generalize COPs. The learned encoder on several problems can directly perform comparably on new problems to models trained from scratch, suggesting its potential to support developing the foundational model for combinatorial optimization.

## 1 INTRODUCTION

Combinatorial optimization problems (COPs), which involve optimizing discrete variables under specific objectives, are fundamental and serve widespread practical applications (Hong et al., 2010; Mironov & Zhang, 2006; Ganzinger et al., 2004). Despite decades of progress, the inherent computational complexity of COPs still demands substantial expert effort to craft effective heuristics. Recently, advances in machine learning (ML) have demonstrated the potential to automatically learn solving heuristics from data, improving solution quality and speed when the problem instances fall within certain distributions (Bengio et al., 2021; Kool et al., 2018; Joshi et al., 2019; Kwon et al., 2020; Sun & Yang, 2023; Li et al., 2023). However, these gains often remain isolated: most learned solvers are trained per problem and struggle to transfer knowledge across tasks, even when those tasks share striking structural similarities.

Despite their complexity, many COPs share common structures, exhibiting similarities in optimization objectives, decision variables, or constraints (Kool et al., 2018; Kwon et al., 2020), and are often connected by polynomial-time reductions (Kleinberg & Tardos, 2006). The presence of these commonalities suggests that techniques designed for one COP may have broader relevance, offering the potential for more generalizable and adaptable solutions. Historically, classical heuristics (Papadimitriou & Steiglitz, 2013; Dorigo et al., 2006) have demonstrated this versatility, proving effective across various related problems with minimal adaptation. This observation raises the possibility that machine learning models, when trained on diverse COPs, could similarly capture shared knowledge across tasks, learning generalized problem-solving strategies. Such an approach could enable these models to perform well on new, unseen problems with minimal task-specific fine-tuning.

This paper tries to learn generalized solving models that are effective across multiple COPs. One crucial characteristic of COPs is their deterministic optimization objectives that directly evaluate solving

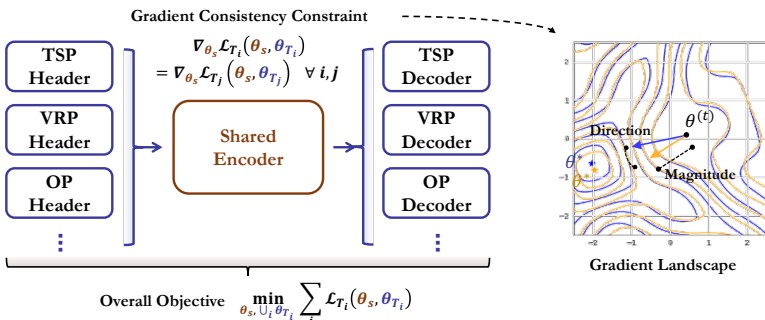

Figure 1: The GCNCO framework utilizes light-parameterized headers and decoders to handle problem-specific inputs and outputs, respectively, and enforces a gradient consistency constraint on the heavy-parameterized shared encoder to extract generalized strategies. The gradient consistency constraint involves homogenizing the gradients of different problems regarding direction and magnitude.

performance, enabling the training loss to align precisely with the testing evaluation metric through reinforcement learning. By incorporating objectives into the training phase, the gradients explicitly reflect how updates to the models internal strategy representation improve solving performance. This objective correspondence motivates the use of gradient-based methods to regularize the learning process. Furthermore, while COPs vary in form, many NP-hard problems share polynomial-time reducibility (e.g., TSP reduces to VRP). We hypothesize that different COPs may exhibit structural similarities in a unified latent space, allowing their optimization behaviors to be homogenized.

As shown in Fig. 1, we introduce gradient-consistent neural combinatorial optimization (GCNCO) to enforce the gradient consistency of the optimization across tasks and promote the learning of generalizable strategies. At its core, GCNCO employs a header-encoder-decoder architecture that processes diverse problems in a unified manner. The header and decoder handle problem-specific inputs and outputs, and the encoder is designed to learn generalized strategies within a shared latent space. We enforce gradient consistency on the encoder so that updates induced by different COP objectives point in similar directions and have comparable magnitudes. When a models optimization trajectory, reflected in both gradient directions and magnitudes, is consistent across tasks, it indicates that optimizing for one task simultaneously improves the models efficacy on others, suggesting that the model is acquiring a shared, underlying solution strategy applicable to various COPs.

Specifically, we propose to align both gradient directions and magnitudes of the encoder during training through task-specific feature rotation matrices and loss weights, which steer the encoders gradients. The feature rotation matrices rotate the input features and output features of the encoder to homogenize its optimization landscape and thus adjust the gradient direction, while the loss weights modulate the gradient magnitude. The whole framework follows a bi-level optimization process where the original models are optimized through the solving objective of different problems by reinforcement learning, and the additional rotation matrices and loss weights are optimized through a gradient-consistent loss reflecting the gradient homogenization.

Our experiments cover six combinatorial optimization problems: Travelling Salesman Problem (TSP), Vehicle Routing Problem (VRP), Split Delivery VRP (SDVRP), Orienteering Problem (OP), Prize Collecting TSP (PCTSP), and Stochastic PCTSP (SPCTSP). We analyze the relationships between tasks by studying the gradient alignment across problems. We then demonstrate the effectiveness of GCNCO in learning a general solving strategy by evaluating its performance on both training tasks and new, unseen problems under zero-shot evaluation and fine-tuning settings. Remarkably, our learned encoder on new problems achieves results comparable to models trained from scratch, highlighting the potential of our approach to serve as a foundational model for combinatorial optimization.

## 2 RELATED WORK

**Neural Combinatorial Optimization.** Recent advancements in machine learning for combinatorial optimization (CO) include constructive and improvement-based approaches. Constructive methods, including autoregressive techniques (Khalil et al., 2017; Kool et al., 2018; Kwon et al., 2020; Hottung et al., 2021; Kim et al., 2022), sequentially determine decision variables to build complete solutions, while non-autoregressive strategies (Joshi et al., 2019; Fu et al., 2021; Geisler et al., 2022; Qiu

et al., 2022; Sun & Yang, 2023) generate soft-constrained solutions in a single pass and then enforce post-processing for feasibility. In contrast, improvement-based solvers (d O Costa et al., 2020; Wu et al., 2021; Chen & Tian, 2019; Li et al., 2021; Hou et al., 2023) iteratively refine existing solutions using local search operators to optimize the objective. Recently, generative modeling has shown promise in CO, e.g., diffusion models (Sun & Yang, 2023; Li et al., 2023), GFlowNets (Zhang et al., 2023), and consistency models (Li et al., 2024), framing problem-solving as a conditional generation task to learn solution distributions tailored to specific instances.

**Multi-Task learning**. Multi-Task Learning (MTL) seeks to address multiple tasks by training a unified model that captures shared insights across these tasks. Studies have tackled MTL from different angles, such as striking a balance in the loss functions of various tasks (Mao et al., 2021; Dai et al., 2023), developing mechanisms for module sharing (Javaloy & Valera, 2021), and leveraging meta-learning (Wang et al., 2021). To improve MTL efficiency and reduce the detrimental effects of negative transfer (Jiang et al., 2024), some research has turned to task grouping strategies (Fifty et al., 2021), to discern task relationships and facilitate learning within these groups to minimize negative transfer between conflicting tasks. MTL has found widespread application in many fields (Allenspach et al., 2024; Zhang et al., 2024). However, there is little research on applying MTL to COPs.

**MTL for CO.** Unified models capable of simultaneously addressing multiple COPs is still in early stages. Wang et al. (2025) proposes a multi-armed bandit approach to solve multiple COPs by alternating optimizing different problems. For specialized problems like Vehicle Routing Problems (VRPs), Lin et al. (2024) suggest training a shared backbone model, which can then be fine-tuned for different VRP variants using linear projections. Additionally, Zhou et al. (2024); Liu et al. (2024); Berto et al. (2024) apply attribute composition to achieve (zero-shot) generalization across various VRP variants. While these approaches treat a set of problem variants as extensions of a single-task model with varying attributes, this paper explores learning generalized strategies across different problems, aiming to prune the single-task model for its generalized part rather than expanding it.

# 3 PRELIMINARIES

Following Karalias & Loukas (2020); Wang et al. (2022), we define $\mathcal{G}$ as the set of CO problem instances represented by graphs $G(V, E) \in \mathcal{G}$, where $V$ and $E$ denote the nodes and edges, respectively. CO problems can be classified into two types: edge-decision problems, which involve selecting edges, and node-decision problems, which involve selecting nodes. Let $\mathbf{x} \in \{0, 1\}^N$ be the optimization variable, where each entry indicates inclusion in the solution. For edge-decision problems, $N = n^2$, and $\mathbf{x}_{i,j}$ denotes whether edge $E_{i,j}$ is included. For node-decision problems, $N = n$, and $\mathbf{x}_i$ denotes whether node $V_i$ is included. The feasible set $\Omega$ consists of all $\mathbf{x}$ that satisfy the problems constraints. A CO problem on $G$ seeks a feasible $\mathbf{x}$ that minimizes the objective $l(\cdot; G) : \{0, 1\}^N \to \mathbb{R}_{\geq 0}$:

$$\min_{\mathbf{x} \in \{0,1\}^N} l(\mathbf{x}; G) \quad \text{subject to} \quad \mathbf{x} \in \Omega \tag{1}$$

Different COPs exhibit variations in their loss functions and constraints. In this paper, we consider several such problems: TSP, an edge-decision problem where the goal is to find the shortest tour that visits each node once and returns to the starting point; CVRP, which involves determining vehicle routes starting and ending at a depot, while ensuring the total demand on each route does not exceed vehicle capacity; SDVRP, which extends VRP by allowing deliveries to be split across multiple routes; OP, a variant of TSP aimed at maximizing the total prize collected from visited nodes under a distance constraint, without requiring every node to be visited; PCTSP, where each node has both a prize and a penalty for being unvisited, and the goal is to collect a minimum total prize while minimizing the tour length and unvisited-node penalties; and SPCTSP, which introduces uncertainty by revealing the actual prize only when a node is visited, while the expected prize is known beforehand.

# 4 GRADIENT-CONSISTENT NEURAL COMBINATORIAL OPTIMIZATION

## 4.1 OPTIMZIATION FOR GENERALIZED STRATEGIES

Consider a set of tasks $\{T_i\}_{i=1}^k$, where the goal is to learn a neural network solver $S_\theta$ that maps $X_i$ to $Y_i$, i.e., $S_\theta : X_i \to Y_i$. The primary aim is to learn a generalizable solution strategy that captures the shared information across multiple tasks while also maintaining task-specific capabilities. As a result, this model processes the learned strategies being effective across multiple tasks, which may

potentially be effective for new, unseen tasks. This model can serve as a pre-trained model, where its performance on a specific task can be optimized through tuning.

To achieve this, we structure the model with a shared network $\theta_s$ responsible for learning generalized strategies, supplemented by task-specific modules $\theta_{T_i}$ designed to handle the unique input and output form of each task. The complete model is represented as $\{\theta_s, \theta_{T_i}, \ldots, \theta_{T_k}\}$, where for a particular task $T_i$, the parameters involved are $\{\theta_s, \theta_{T_i}\}$. The shared network $\theta_s$ captures knowledge common across tasks, while the task-specific modules $\theta_{T_i}$ ensure sufficient specialization for each problem.

The optimization objective for learning these generalized strategies is to ensure that, with the support of the task-specific modules $\theta_{T_i}$, the relationship $P(X_i \rightarrow Y_i)$ remains consistent and stable across tasks, reflecting a potential unified problem-solving strategy. This can be expressed as:

$$P(X_i \rightarrow Y_i | \theta_{T_i}) = P(X_j \rightarrow Y_j | \theta_{T_j}), \forall i, j \tag{2}$$

Then, there exists a neural model $\theta_s$ corresponding to this mapping relation, and this consistency in problem-solving strategies across tasks ensures that updates to $\theta_s$ lead to simultaneous improvements. In other words, the shared parameters $\theta_s$ should contribute to all tasks in a stable and coherent manner.

A key characteristic of COPs is their deterministic optimization objectives, which enables the training loss to align precisely with the evaluation metric (via reinforcement learning), further ensuring that gradients during training explicitly reflect improvements in solving performance. Thus, we derive a gradient consistency constraint during the optimization process to ensure coherent improvements from $\theta_s$ updates across COPs. This constraint ensures that the gradient directions and magnitudes for $\theta_s$ across tasks remain aligned, thereby guiding the model toward a generalized solution:

$$\min_{\theta_s, \bigcup_i \theta_{T_i}} \sum_i \mathcal{L}_{T_i}(\theta_s, \theta_{T_i})$$
$$s.t. \quad \nabla_{\theta_s} \mathcal{L}_{T_i}(\theta_s, \theta_{T_i}) = \nabla_{\theta_s} \mathcal{L}_{T_j}(\theta_s, \theta_{T_j}), \forall i, j \tag{3}$$

Therefore, gradient alignment ensures the shared network learns strategies beneficial across all tasks, enhancing generalization and robustness in the learned solution strategies.

## 4.2 THE GENERAL MODEL DESIGN

To facilitate direct performance correspondence and simplify constraint handling, we directly take the problem objectives as the training loss through reinforcement learning and use a sequential decision-making approach, where a point is selected at each step to be integrated into the current partial solution, ultimately creating a complete solution. This method ensures that constraints are met at each step, thus addressing the differences in constraints across multiple problems in a unified manner. Without loss of generality, taking TSP as an example, the solution sequence $\pi = (\pi_1, \pi_2, \ldots, \pi_n)$ is treated as a permutation of nodes, depending on the problem type. The model defines a stochastic policy $p_\theta(\pi|s)$ for selecting the sequence solution $\pi$ given a problem instance $G$:

$$p_\theta(\pi|G) = \prod_{t=1}^{n} p_\theta(\pi_t | G, \pi_{1:t-1}), \tag{4}$$

where $\pi_t$ represents the decision at step $t$, conditioned on the current partial solution $\pi_{1:t-1}$ and problem instance $s$. The policy $p_\theta$ is parameterized by $\theta$, typically learned through a graph neural network or attention mechanism. The probability distribution $p_\theta(\pi_t | G, \pi_{1:t-1})$ determines the selection of the next node based on the current graph context and previously selected components. The model can be directly optimized through the expectation of the given problem objective $\mathcal{L}(\theta|G) = \mathbb{E}_{p_\theta(\pi|G)}[L(\pi)]$. We optimize $\mathcal{L}$ by gradient descent, using the REINFORCE (Williams, 1992) gradient estimator with baseline $b(G)$ following Kool et al. (2018); Kwon et al. (2020):

$$\nabla \mathcal{L}(\theta|G) = \mathbb{E}_{p_\theta(\pi|G)} \left[ (L(\pi) - b(G)) \nabla \log p_\theta(\pi|G) \right]. \tag{5}$$

Many NP-hard COPs exhibit structural similarities due to polynomial-time reducibility (e.g., TSP to VRP), suggesting the existence of a potential unified latent space where solution strategies can be homogenized. To capitalize on these structural commonalities and enforce gradient consistency within a coherent latent space, we adopt separate headers and decoders for each problem to handle problem-specific inputs and outputs. The entire model follows a header-encoder-decoder framework, where the headers and decoders consist of a single layer each, while the encoder, with significantly larger parameters, is designed to learn the core solving strategy.

Figure 2: Representation flow of the GCNCO framework. $f_{h_i}, f_\theta, f_{d_i}$ denote the mappings of the header, encoder, and decoder, respectively. $\mathbf{R}'_i, \mathbf{R}_i$ denote the introduced rotation matrices that correspond to the input features and the output features.

The headers preprocess raw input from various COPs, converting data like coordinates, demands, rewards, and penalties into unified vector representations. The shared encoder follows Transformer (Vaswani, 2017) principles, mapping inputs to $Q, K, V$ via linear layers $W_q, W_k, W_v$ then applying multi-head attention. The output is combined with the input via residual connections and layer normalization, before being processed by a feed-forward network to extract task-specific features. The decoding process differs from the Transformer decoder, using a combination of multi-head and single-head attention. In the final step, attention scores and embedding vectors are generated based on the current problem state (e.g., current node, remaining capacity) using multi-head attention, followed by single-head attention to compute action probabilities, simplifying the action selection process and outputting the probability distribution, which is more suitable for optimization tasks.

## 4.3 GRADIENT HOMOGENIZATION

To enable the encoder to learn a unified strategy effective across all tasks, corresponding to the relationship $P(X_i \to Y_i | \theta_{T_i})$, its gradient-based optimization behavior should be consistent across tasks within a shared latent space. Since directly estimating the full gradient field is intractable, we instead standardize the encoders gradients along the optimization trajectory. Specifically, we follow Javaloy & Valera (2021) here to homogenize the gradients of the encoder in a header-encoder-decoder architecture regarding gradient directions and magnitudes. However, unlike traditional multi-task learning settings with a single shared feature stream as in Javaloy & Valera (2021), our design introduces task-specific input and output transformations, invalidating the assumption of identical feature flows. We therefore adapt the gradient-alignment mechanism to this architecture via a bi-level scheme that explicitly accounts for the encoders input and output features, as detailed below.

**Feature-Level Gradients.** Directly homogenizing the gradients of encoder parameters across COPs incurs significant computational overhead. Thus, we resort to homogenizing gradients of output features of the encoder. Since different headers may process the same instance (with slight variations, e.g., additional constraints) into distinct representations, we denote the header output feature for task $T_i$ as $\mathbf{w}_i$ and the encoder output feature for task $T_i$ as $\mathbf{z}_i$. The representation flow is shown in Fig. 2. Traditionally, when the shared encoder receives identical data inputs, the encoder output feature is shared, and the gradient of the loss $L_{T_i}$ with respect to the encoder parameters $\theta$ can be expressed as $\nabla_\theta L_{T_i} = \nabla_\theta \mathbf{z} \cdot \nabla_{\mathbf{z}} L_{T_i}$, where $\nabla_\theta \mathbf{z}$ is shared across tasks, and $\nabla_{\mathbf{z}} L_{T_i}$ captures task-specific variations (Javaloy & Valera, 2021). However, in our header-encoder-decoder framework, inconsistent $\mathbf{w}_i$ induce distribution shifts in encoder inputs, undermining this approximation. Without constraining the gradient directions of $\mathbf{w}_i$, different headers may project the original input $x$ into significantly divergent latent spaces (e.g., $\mathbf{w}_1$ and $\mathbf{w}_2$ may exhibit distribution shifts). This undermines the effectiveness of the feature-level gradient approximation.

To this end, we additionally enforce consistency in the gradient directions of $\nabla_{\mathbf{w}_i} L_{T_i}$ to implicitly align the input feature spaces, ensuring that the encoder receives inputs with similar optimization-driven feature evolution patterns. This reduces the encoder's burden of adapting to task-specific noise and allows it to focus on capturing cross-task commonalities. If the input gradients of tasks $i$ and $j$ satisfy $\nabla_{\mathbf{w}_i} L_{T_i} \propto \nabla_{\mathbf{w}_j} L_{T_j}$, The header networks adjust mappings in a consistent direction. Consequently, although the shared encoder receives input features $\mathbf{w}_i$ from different tasks, their optimization-driven feature changes follow a similar pattern. This alignment enables the encoder parameters $\theta$ to focus on learning shared strategies rather than overfitting task-specific variations.

**Gradient Magnitude Homogenization.** We adopt a hyperparameter-free method normalizing gradient magnitudes across tasks, following Normalized Gradient Descent (Cortés, 2006)

and Javaloy & Valera (2021). Let the feature-level gradients of the task $T_i$ for the $k$-th data point be denoted as $\mathbf{g}_{T_i,k} = \nabla_{\mathbf{z}_k}\mathcal{L}_{T_i}(\mathbf{z}_k)$, and the batch gradient for task $T_i$ is represented as $\mathbf{G}_{T_i}^\top := [\mathbf{g}_{T_i,1}, \mathbf{g}_{T_i,2}, \ldots, \mathbf{g}_{T_i,B}]$, where $B$ is the batch size. Then, we can intuitively rescale the gradients of different problems through the normalization: $\mathbf{U}_{T_i} = \frac{\mathbf{G}_{T_i}}{\|\mathbf{G}_{T_i}\|}, \forall i$, which balances the gradient magnitudes across different tasks to gradient units. Denoting the common scalar magnitude for all task gradients denoted as $C$, so the final gradients become $C\mathbf{U}_{T_i}$. We define $C$ as a convex combination of task-specific gradient magnitudes at $t$:

$$C := \sum_i \alpha_{T_i}\|\mathbf{G}_{T_i}\|, \tag{6}$$

where the weights $\alpha_{T_i}$ (which sum to 1) reflect each tasks convergence speed and are defined as:

$$\alpha_{T_i} = \frac{\|\mathbf{G}_{T_i}\|/\|\mathbf{G}_{T_i}^0\|}{\sum_j \|\mathbf{G}_{T_j}\|/\|\mathbf{G}_{T_j}^0\|}, \tag{7}$$

where $\mathbf{G}_{T_i}^0$ is the initial gradient for task $T_i$ (i.e., at train iteration $t = 0$). This arrangement dynamically adjusts the scaling based on the convergence speed of each task, allowing tasks with slower convergence larger step sizes, while those that converge more quickly receive smaller updates. As a result, the optimization of the solving objectives can be adjusted to a weighted loss minimization process, where the loss weight for problem $T_i$ is $\frac{C}{\|\mathbf{G}_{T_i}\|}$. The optimization is then modified as:

$$\min_{\theta_s, \bigcup_i \theta_{T_i}} \sum_i \frac{C}{\|\mathbf{G}_{T_i}\|}\mathcal{L}_{T_i}(\theta_s, \theta_{T_i}). \tag{8}$$

**Gradient Direction Homogenization.** To homogenize the gradient directions, we follow Javaloy & Valera (2021) and introduce problem-specific rotation matrices on the hidden representations to rotate the optimization landscape, thereby adjusting the optimization directions for gradient consistency. However, since the inputs to the shared encoder are derived from different headers, in addition to constraining $\nabla_{\mathbf{z}_i}L_{T_i}$, we also enforce consistency in the gradient directions of $\nabla_{\mathbf{w}_i}L_{T_i}$ to implicitly align the input feature spaces of the encoder. For each task $T_i$, we introduce rotation matrices $\mathbf{R}_i', \mathbf{R}_i \in SO(d)$ to align the input gradients and output gradients of the shared encoder with a unified direction. Then optimizing the loss calculated with the rotated latent representation $\mathbf{r}_{\mathbf{w}_i} = \mathbf{R}_i'\mathbf{w}_i$ and $\mathbf{r}_{\mathbf{z}_i} = \mathbf{R}_i\mathbf{z}_i$ can lead to the rotation of the optimization landscapes (Soltanolkotabi et al., 2018), thereby homogenize the gradients across tasks.

Since the rotation matrices introduce additional parameters that only affect the gradient directions for different tasks, we optimize them by minimizing the conflict between the task-specific gradients of the encoder by aligning them toward a common direction. This is achieved by maximizing the cosine similarity of task gradients. The objective for optimizing the rotation matrices is to minimize:

$$\mathcal{L}_{\text{rot}}(\mathbf{R}_1, \mathbf{R}_2, \cdots, \mathbf{R}_k) = -\sum_i \langle \mathbf{R}_i^\top \nabla_{\mathbf{r}_{\mathbf{z}_i}}\mathcal{L}_{T_i}, \mathbb{E}_j(\mathbf{U}_{T_j}) \rangle$$

$$\mathcal{L}_{\text{rot}}'(\mathbf{R}_1', \mathbf{R}_2', \cdots, \mathbf{R}_k') = -\sum_i \langle \mathbf{R}_i'^\top \nabla_{\mathbf{r}_{\mathbf{w}_i}}\mathcal{L}_{T_i}, \mathbb{E}_j(\mathbf{U}_{T_j}') \rangle \tag{9}$$

where $\mathbf{U}_{T_j}$ and $\mathbf{U}_{T_j}'$ denotes the normalized gradients of the output and input feature of the encoder for problem $T_j$. $\mathbb{E}_j(\mathbf{U}_{T_j})$ and $\mathbb{E}_j(\mathbf{U}_{T_j})$ is the target direction that all task gradients should point toward, which we select as the average normalized gradient direction across all tasks.

### 4.4 OVERVIEW

Based on the header-encoder-decoder architecture, when optimizing the objective function, we additionally attach rotation matrices for different tasks to transform the input and output features of the encoder. These transformed features are decoded via problem-specific decoders to obtain the results, and the model is optimized by Eq. 8. Simultaneously, we update the parameters of the rotation matrix via Eq. 9, optimizing the rotation matrix to align gradient directions consistently across tasks.

In this framework, the model is trained simultaneously on data from multiple COPs, encouraging the encoder to capture shared problem-solving strategies that transcend individual tasks. These strategies are designed to generalize to unseen problems, reducing the need for extensive retraining when

encountering new tasks. After the initial training phase, when applied to a new problem, the encoder can remain frozen, leveraging its pre-learned strategies to provide strong initial performance. In this scenario, only the lightweight, problem-specific header and decoder need to be retrained, significantly reducing computational overhead. Alternatively, the encoder can be fine-tuned on new tasks to further specialize its problem-solving capabilities, potentially achieving higher performance.

## 5 EXPERIMENTS

This section provide empirical validations by analyzing the relationship between different COPs through gradient similarity, assessing the in-distribution solving performance, and investigating the model's generalization ability across tasks, both in a zero-shot setting and with fine-tuning.

### 5.1 EXPERIMENTAL SETUP

**Datasets.** The node coordinates of different problems are uniformly sampled within the unit square $[0, 1]^2$, with additional constraints identified by sampling rules specified in Appendix C. All settings mentioned above are a standard procedure as adopted in Kool et al. (2018); Hottung et al. (2021); Sun & Yang (2023); Li et al. (2023). We experiment on the problem scales of 20 and 50 for every COP.

**Baselines.** We compare our model to the baseline from Wang et al. (2025), which proposes a multi-task learning framework based on the Multi-Armed Bandit (MAB) for dynamically selecting training tasks for COPs, using identical experimental settings. It employs the MAB algorithm to select the task to be optimized in each round, constructs reward signals through loss decomposition and guides the task selection strategy. We also include AM models (Kool et al., 2018), which do not incorporate multi-task considerations, for additional comparison. To further evaluate performance, we benchmark against several mainstream heuristic solvers: LKH (Helsgaun, 2017), which dynamically adjusts edge exchanges in paths to eliminate crossings and approximate the optimal solution; Gurobi (Gurobi Optimization, 2020), which combines exact algorithms like branch-and-bound and cutting-plane methods with preprocessing and heuristics to solve COPs; and Compass (Kobeaga et al., 2018), which uses a directional search-based heuristic to balance local optimization with global search capabilities in combinatorial optimization.

**Metrics.** Following Kool et al. (2018); Joshi et al. (2019); Sun & Yang (2023); Li et al. (2023), we adopt two metrics: 1) Obj: the value of the objective function, which is the actual result of the optimization. In a minimization problem, the smaller the objection value, the better the solution, while in a maximization problem, a larger objection value is preferred. 2) Gap: the closeness of the solution to the optimal solution, representing the difference between the current solution and the known reference solution or the theoretical bound.

### 5.2 PROBLEM RELATIONSHIP AND GENERALIZATION MEASURE

In this experiment, we aim to analyze the underlying relationships between different CO problems by examining the similarity of their gradients during optimization, which can also serve as an indicator of how generalized the learned model is. We train the model simultaneously on classic routing problems, including TSP variants and VRP variants, using the rotation matrices to enforce consistency in the gradient directions across problems. By monitoring the gradients, we observe the degree of alignment between the problems, which reveals the extent of shared structure in the optimization process. The gradient similarity is computed using the cosine similarity of the gradients across tasks, demonstrating how related the optimization landscapes of different CO problems are.

**Results.** Fig. 3 (a) shows the trend of task correlation reflected by the cosine similarities of the encoder gradients across problem pairs, compared to the direct baseline MCOMAB (Wang et al., 2025). It shows the training process in fine detail, where our method not only outperforms the baseline in all task pairs but also converges in the later stages of training, demonstrating its potential to learn inter-task correlations. Fig. 3 (b) presents the comparison of the problem correlation heatmaps between our method and the baseline method. The problem similarities learned from different models are measured by the average cosine similarities of the encoder gradients across the training process. GCNCO shows improvement across major tasks, which enables learning more inter-task correlations during the training process. On the other hand, we discover a higher correlation between the TSP and VRP variants, while it is more challenging to homogenize OP and other routing problems.

### 5.3 IN-DISTRIBUTION SOLVING PERFORMANCE

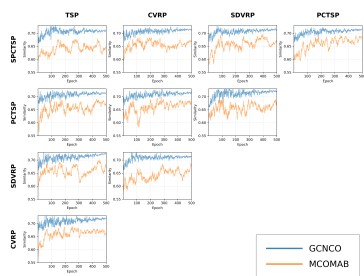
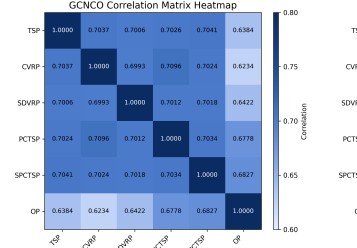
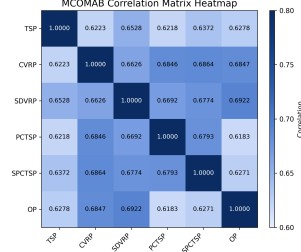

(a) Training-time cosine similarities    (b) Correlation heatmap of average cosine similarities

Figure 3: (a) The cosine similarities of the encoder gradients in the training process across problem pairs. (b) The correlation heatmaps of average cosine similarities of the encoder gradients across problem pairs for GCNCO and MCOMAB, where GCNCO achieves clearly higher similarity scores.

This experiment evaluates the model's in-distribution solving performance on multiple CO tasks. We jointly train the model on the six CO tasks and measure its performance on these tasks. The encoder is designed to capture shared strategies, while problem-specific rotation matrices and decoders handle individual task outputs. After the joint training, we then perform additional fine-tuning on each task individually to assess whether further improvements can be obtained. The performance is evaluated based on standard metrics such as solution quality and computational efficiency.

**Results.** The in-distribution performance reflects how well the model handles tasks on which it has been trained. Since learning generalized strategies may reduce the model's focus on task-specific structures, we further fine-tune the pre-trained models on individual tasks. Table 4 shows the superiority of GCNCO. The performance gains of GCNCO compared to MCOMAB are especially consistent on 50-node instances.

Figure 4: In-distribution solving performance. * indicates reference for computing gaps.

| PROBLEM | ALGO. | $n = 20$ | | $n = 50$ | |
|---|---|---|---|---|---|
| | | Obj. | Gap | Obj. | Gap |
| TSP | LKH3 | $3.84^*$ | 0.00% | $5.70^*$ | 0.00% |
| | AM | 3.85 | 0.08% | 5.71 | 0.35% |
| | MCOMAB | 3.84 | 0.04% | 5.70 | 0.30% |
| | GCNCO | 3.84 | **0.03%** | 5.70 | **0.25%** |
| CVRP | LKH3 | $6.13^*$ | 0.00% | $10.38^*$ | 0.00% |
| | AM | 6.19 | 0.93% | 10.61 | 2.20% |
| | MCOMAB | 6.18 | 0.86% | 10.59 | 2.15% |
| | GCNCO | 6.17 | **0.68%** | 10.56 | **1.77%** |
| SDVRP | AM | 6.25 | 3.96% | $10.59^*$ | **0.00%** |
| | MCOMAB | $5.96^*$ | **0.00%** | 10.81 | 2.13% |
| | GCNCO | 6.03 | 1.15% | 10.81 | 2.11% |
| PCTSP | Gurobi | $3.13^*$ | 0.00% | $4.48^*$ | 0.00% |
| | AM | 3.19 | **1.88%** | 4.60 | 2.74% |
| | MCOMAB | 3.34 | 6.79% | 4.56 | 1.83% |
| | GCNCO | 3.24 | 3.67% | 4.53 | **1.32%** |
| SPCTSP | AM | 3.26 | 0.07% | 4.64 | 0.69% |
| | MCOMAB | 3.25 | 0.03% | 4.67 | 1.31% |
| | GCNCO | $3.24^*$ | **0.00%** | $4.61^*$ | **0.00%** |
| OP | Compass | $5.37^*$ | 0.00% | $16.17^*$ | 0.00% |
| | AM | 5.30 | 1.56% | 16.08 | 0.53% |
| | MCOMAB | 5.34 | 0.70% | 16.08 | 0.58% |
| | GCNCO | 5.34 | **0.43%** | 16.09 | **0.46%** |

### 5.4 CROSS-PROBLEM GENERALIZATION

One of the key strengths of GCNCO lies in its ability to generalize across different CO problems. To evaluate this, we conduct experiments that assess the model's cross-task generalization under two settings: task-specific fine-tuning and zero-shot generalization with the pretrained encoder.

#### 5.4.1 ZERO-SHOT CROSS-PROBLEM GENERALIZATION

We investigate the model's ability to generalize to new tasks without additional training. We freeze the encoder after training it on five CO tasks and directly apply it to solve the sixth, previously unseen task. This zero-shot setting allows us to assess the extent to which the learned solving strategies can be transferred to new tasks. We evaluate the model's performance on the new task and compare it to the performance of models trained from scratch on the same task, highlighting the effectiveness of the shared encoder in transferring knowledge across problems.

**Results.** We freeze the encoder parameters that has already been trained on five tasks, and compare its performance with a model trained from scratch using AM (Kool et al., 2018) and MCOMAB (Wang et al., 2025), all trained under identical conditions. Metrics are evaluated on the previously unseen problem to assess generalization. As shown in Table 5, even without task-specific training, the frozen encoder outperforms baselines specifically trained or fine-tuned on the target task, highlighting the ability of multi-task training to capture commonalities across different CO tasks.

#### 5.4.2 FINETUNING FOR CROSS-PROBLEM GENERALIZATION

To further explore the generalization ability of the model, we evaluate the performance of fine-tuning the pre-trained encoder on the new task. In this setting, after initially freezing the encoder, we perform additional fine-tuning on the new task-specific data. This quantifies how well the model adapts to new tasks when a small amount of task-specific data is available. The results of this experiment are compared to the zero-shot setting and the baseline models trained from scratch, demonstrating the encoder's flexibility in handling both new and previously seen tasks.

**Results.** We further fine-tune the learned encoder and compare its performance with AM (Kool et al., 2018) trained from scratch and fine-tuned MCOMAB (Wang et al., 2025). Table 5 shows that, under the same training setup with five CO tasks, our model achieves a larger gap improvement on unseen tasks after fine-tuning, with results numerically closer to the optimal solutions provided by heuristic solvers. This indicates that fine-tuning on the target task complements the model's ability to incorporate task-specific information and improve performance, highlighting its

Figure 5: Cross-problem generalization with models trained on five problems and evaluated on the left-out problem. * indicates the baseline for computing the performance gap.

| PROBLEM | ALGO. | $n = 20$ | | $n = 50$ | |
|---|---|---|---|---|---|
| | | Obj. | Gap | Obj. | Gap |
| TSP | LKH3 | 3.84* | 0.00% | 5.70* | 0.00% |
| | AM | 3.85 | 0.09% | 5.71 | 0.38% |
| | MCOMAB-ftune | 3.84 | 0.05% | 5.71 | 0.35% |
| | GCNCO-frozen | 3.84 | 0.05% | 5.71 | 0.33% |
| | GCNCO-ftune | 3.84 | **0.04%** | 5.70 | **0.29%** |
| CVRP | LKH3 | 6.13* | 0.00% | 10.38* | 0.00% |
| | AM | 6.19 | 0.91% | 10.61 | 2.24% |
| | MCOMAB-ftune | 6.19 | 0.93% | 10.60 | 2.09% |
| | GCNCO-frozen | 6.18 | 0.82% | 10.58 | 1.89% |
| | GCNCO-ftune | 6.17 | **0.71%** | 10.55 | **1.64%** |
| SDVRP | AM | 6.25 | 5.22% | 10.59* | **0.00%** |
| | MCOMAB-ftune | 6.06 | 2.02% | 11.73 | 10.71% |
| | GCNCO-frozen | 6.07 | 2.19% | 10.65 | 0.60% |
| | GCNCO-ftune | 5.94* | **0.00%** | 10.60 | 0.08% |
| PCTSP | Gurobi (10s) | 3.13* | 0.00% | 4.48* | 0.00% |
| | AM | 3.19 | 1.82% | 4.58 | 2.24% |
| | MCOMAB-ftune | 3.63 | 14.01% | 5.34 | 19.29% |
| | GCNCO-frozen | 3.44 | 9.87% | 4.62 | 3.19% |
| | GCNCO-ftune | 3.18 | **1.56%** | 4.53 | **1.31%** |
| SPCTSP | AM | 3.26 | 1.62% | 4.64 | 3.40% |
| | MCOMAB-ftune | 3.33 | 3.77% | 4.68 | 4.29% |
| | GCNCO-frozen | 3.28 | 2.30% | 4.54 | 1.14% |
| | GCNCO-ftune | 3.21* | **0.00%** | 4.49* | **0.00%** |
| OP | Compass | 5.37* | 0.00% | 16.17* | 0.00% |
| | AM | 5.31 | 1.16% | 16.08 | 0.54% |
| | MCOMAB-ftune | 5.34 | 0.51% | 16.10 | **0.43%** |
| | GCNCO-frozen | 5.34 | 0.55% | 16.08 | 0.55% |
| | GCNCO-ftune | 5.34 | **0.43%** | 16.10 | **0.43%** |

potential as a foundation model for various tasks. Compared to the baselines, the task-specific fine-tuning process yields clear gains over the frozen encoder, and in the full training, GCNCO substantially outperforms the MCOMAB baseline.

## 5.5 ABLATION STUDY OF GRADIENT MAGNITUDE AND DIRECTION HOMOGENIZATION

To assess the core componentsgradient magnitude homogenization (a) and gradient direction homogenization (b), we perform ablation studies in our full fine-tuning pipeline. Models are trained on five COPs, then fine-tuned on the held-out problem, with TSP and CVRP used as examples. We compare variants without either technique, with only (a), with only (b), and with both (a) and (b). Each technique

Figure 6: Ablation study on gradient magnitude and direction homogenization. Models trained on five COPs, fine-tuned and tested on TSP & CVRP.

| ALGORITHM | TSP | | CVRP | |
|---|---|---|---|---|
| | Obj. | Gap | Obj. | Gap |
| LKH3 | 3.848* | 0.00% | 6.134* | 0.00% |
| w/o mag. homog., w/o dir. homog. | 3.851 | 0.08% | 6.191 | 0.93% |
| w/ mag. homog, w/o dir. homog. | 3.850 | 0.06% | 6.185 | 0.83% |
| w/o mag. homog., w/ dir. homog. | 3.849 | 0.06% | 6.188 | 0.88% |
| w/ mag. homog., w/ dir. homog. | **3.849** | **0.04%** | **6.176** | **0.71%** |

boosts cross-problem generalization, while their combination yields the largest gains, confirming the need to align both gradient magnitudes and directions.

**Results.** Table 6 shows that both magnitude and direction homogenization individually improve over the no-homogenization baseline, with comparable gains. Crucially, applying both together yields the largest gap reduction on TSP and CVRP, demonstrating that synchronizing gradient magnitudes and directions is key to optimal performance.

## 6 CONCLUSION AND FUTURE WORK

This paper introduces a novel approach for learning shared-solving strategies across diverse COPs by enforcing gradient consistency throughout the optimization process. Built on a header-encoder-decoder architecture, our method explicitly separates problem-specific components (header and decoder) from a generalized, problem-agnostic encoder. By leveraging problem-specific rotation matrices and loss weights, the encoder effectively learns and generalizes solving strategies across multiple COPs. Experiments on six distinct CO problems demonstrate strong in-distribution performance and robust generalization in zero-shot and fine-tuned cross-problem scenarios. Our findings highlight the potential of the learned encoder as a foundation for CO models, enabling adaptation to new problems without retraining from scratch. This opens up avenues for future work, including extending the method to larger data and exploring specific fine-tuning methods.

ETHICS STATEMENT

This study was conducted in accordance with the ICLR Code of Ethics and guided by principles of responsible research. It does not involve human subjects, the use of sensitive personal data, or applications with foreseeable harmful potential. We emphasize transparency, reproducibility, and scientific integrity, with methods, datasets, and evaluations documented and results reported accurately. The authors declare no competing interests or funding-related conflicts that could influence the work. Our goal is to contribute positively to the machine learning community and society by promoting fairness, accessibility, and open scientific inquiry, and we remain committed to proactively identifying and responsibly addressing any unforeseen broader impacts.

REPRODUCIBILITY STATEMENT

We uphold the highest standards of scientific rigor and transparent reporting. To enable faithful reproducibility, we furnish exhaustive documentation of our methodology (Section 4.2), architectural specifications (Subsection 4.3), hyperparameter configurations (Appendix D), inference algorithm (Algorithm 4.2), and dataset curation and configuration (Appendix 5.1) throughout the paper. Code will be made publicly available upon acceptance.

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

APPENDIX

# A    THE USE OF LARGE LANGUAGE MODELS (LLMS)

We employed a large language model exclusively for editorial assistance, i.e., copyediting, rephrasing for clarity, grammar and style consistency, and improving readability and flow. The LLM did not generate ideas, analyses, models, code, experiments, or results, nor did it access proprietary data. All scientific content, methodological choices, and conclusions are the authors own, and all suggested edits were reviewed and verified by the authors before incorporation.

# B    PROBLEM DESCRIPTION AND INSTANCE GENERATION

## B.1    TRAVELING SALESMAN PROBLEM (TSP)

The Traveling Salesman Problem (TSP) aims to find the shortest possible route that visits a set of cities exactly once and returns to the starting city. The objective is to minimize the total travel distance or cost. In the Euclidean version of TSP, the cities are represented as points in a plane, and the distances between cities are the Euclidean distances between their coordinates.

For all TSP instances, the $n$ nodes locations are sampled uniformly at random in the unit square.

**Vehicle Routing Problem (VRP).** The Vehicle Routing Problem (VRP) involves a fleet of vehicles tasked with delivering goods to a set of customers. Each vehicle has a limited capacity, and each customer has a specific demand. The objective is to determine an optimal set of routes for the vehicles to minimize the total travel distance or cost while satisfying constraints such as vehicle capacity and customer demands. The classic VRP assumes that all customers must be visited, and each route starts and ends at a central depot.

In this problem formulation, we introduce a special depot node (indexed as 0) with coordinates $x_0$. Each vehicle has a capacity $D > 0$, and each regular node $i \in \{1, \ldots, n\}$ has a demand $\delta_i$ where $0 < \delta_i \leq D$. Each route begins and ends at the depot, and the total demand of each route must not exceed the vehicles capacity. We denote by $R_j$ the set of node indices assigned to route $j$, ensuring that $\sum_{i \in R_j} \delta_i \leq D$. For simplicity, we normalize the demands by setting $\hat{D} = 1$, using normalized demands $\hat{\delta}_i = \frac{\delta_i}{D}$.

Following Nazari et al. (2018); Kool et al. (2018), we generate instances for $n = 20, 50$, normalizing the demands according to the vehicle capacity. The depot location and node positions are randomly sampled within a unit square, with node demands defined as $\hat{\delta}_i = \frac{\delta_i}{D}$, where $\delta_i$ is the discrete demand sampled uniformly from $\{1, \ldots, 9\}$. Specific demand values for each instance size are $D_{20} = 30$, and $D_{50} = 40$.

**Split Delivery VRP (SDVRP).** The Split Delivery Vehicle Routing Problem (SDVRP) is a variant of the VRP where customer demands may be split across multiple deliveries. This allows a vehicle to make more than one visit to a customer if needed. The objective is to minimize the total distance traveled while satisfying the capacity constraints and meeting the customer demands, which can be split across different routes.

The Split Delivery Vehicle Routing Problem (SDVRP) is a generalization of CVRP, where a node can be visited multiple times, and only a portion of the demand needs to be delivered on each visit. Both CVRP and SDVRP are specified similarly, with instances defined by the number of nodes, the depot location $x_0$, and the normalized demands $\hat{\delta}_i$ for each node $i$. For instance, the problem instances with $n$ nodes and depot at $x_0$ use normalized demands $0 < \hat{\delta}_i \leq 1$, with $i = 1, \ldots, n$.

The instance generation process of SDVRP follows the VRP setting.

**Orienteering Problem (OP).** In the Orienteering Problem (OP), each node is associated with a prize $\rho_i$, and the goal is to maximize the total prize collected from visited nodes while ensuring that the total length of the route does not exceed a maximum length $T$. This differs from the Traveling Salesman Problem (TSP) and the Vehicle Routing Problem (VRP) because visiting each node is optional. To formulate the problem, we introduce a special depot node indexed as 0, located at

coordinates $x_0$. If the model selects the depot, we consider the route to be completed once the depot is revisited. However, to prevent infeasible solutions, we allow visiting the depot only if a return to the depot is still possible within the maximum distance constraint. Its worth noting that visiting the depot is suboptimal if additional nodes can still be visited within the length limit, and we do not enforce that constraint explicitly.

For instance generation, the depot location, along with the $n$ node locations, are uniformly sampled within a unit square. For the prize distribution, the prize is proportional to the distance from the depot, where $\rho_i = 1 + \left| 99 - \frac{d_{oi}}{\max_j d_{oj}} \right|$ and $\hat{\rho}_i = \frac{\rho_i}{100}$, encouraging visits to nodes farther from the depot. The maximum length $T^n$ for instances with $n$ nodes (and the depot) is set to approximately half the length of the average TSP tour for uniformly distributed TSP instances. This design aims to produce challenging instances by maximizing the number of possible node selections, which occurs when $k = \frac{n}{2}$. Fixed maximum lengths are set for different instance sizes: $T^{20} = 2$, and $T^{50} = 3$, adjusted according to the instance size. The length is normalized to align with the unit scale of the node coordinates, ensuring consistency with the normalized prize values.

**Prize Collecting TSP (PCTSP).** The Prize Collecting Traveling Salesman Problem (PCTSP) involves nodes with associated prizes and penalties. The goal is to determine a route that maximizes the total prize collected while minimizing the total travel distance and the penalties for unvisited nodes. In PCTSP, the route is completed once the depot is visited again, but not all nodes need to be visited. The objective is to balance collecting prizes with avoiding penalties, with the total prize constraint being satisfied as a minimum requirement.

For instance generation, the depot and $n$ node locations are randomly sampled in the unit square. The prizes are uniformly distributed with $\rho_i \sim \text{Uniform}(0, 1)$, and the expected total prize for any subset of $\frac{n}{2}$ nodes is $\frac{n}{4}$. The prize for each node is normalized, so $\hat{\rho}_i = \frac{\rho_i}{n}$. In cases where the total prize of visited nodes is less than 1, the prize constraint may be violated, but it is only allowed to return to the depot once all nodes have been visited.

Penalties are also sampled for each node. If the penalties are too small, the selection of nodes is determined almost entirely by the minimum total prize constraint. If the penalties are too large, all nodes are visited, making the total prize constraint irrelevant. To ensure that penalties are meaningful, we sample the penalty values $\beta_i \sim \text{Uniform}(0, 2 \cdot \frac{L_n}{n})$, where $L_n$ is the expected length of the TSP tour with $n$ nodes. This formulation ensures that penalties contribute to the overall objective, balancing the prize collection and penalty minimization.

**Stochastic Prize Collecting TSP (SPCTSP).** The Stochastic Prize Collecting Traveling Salesman Problem (SPCTSP) is an extension of the PCTSP, where the prize collected at each node is uncertain and only revealed when the node is visited. The expected prize $\hat{\rho}_i$ is known in advance, but the actual prize $\rho_i^*$ follows a uniform distribution, with $\rho_i^* \sim \text{Uniform}(0, 2\hat{\rho}_i)$. The objective in SPCTSP remains to maximize the total prize collected while minimizing the total travel distance, but the uncertainty in the actual prizes adds an additional layer of complexity to the problem.

For instance generation, the depot and $n$ node locations are sampled uniformly at random within the unit square. Prizes for the nodes are uniformly distributed, with $\hat{\rho}_i \sim \text{Uniform}(0, 1)$, and the expected total prize for any subset of $\frac{n}{2}$ nodes is $\frac{n}{4}$. The normalized prize for each node is given by $\hat{\rho}_i = \frac{\rho_i}{n}$. Penalties are also assigned, and they are sampled as $\beta_i \sim \text{Uniform}(0, 2 \cdot \frac{L_n}{n})$, where $L_n$ is the expected length of the TSP tour with $n$ nodes. The instance settings follow those used for the PCTSP, with adjusted parameters for the stochastic nature of the prizes.

## C  MODEL ARCHITECTURE

The architecture of the shared encoder follows the transformer architecture adopted in Kool et al. (2018) with 66 attention layers. In decoding, there is also a common handling approach. After obtaining the query vectors for each task, the model computes the multi-head attention output and passes it through a linear layer to integrate the attention results back into the original embedding space. Finally, a single-head attention is used to calculate the score for the current task. This score is scaled and clipped, generating the final probability distribution for the optimal path selection of the given node sequence. The mask is used to handle invalid nodes, ensuring that these nodes are not selected. The designs of the problem-specific headers and decoders are presented below.

## C.1 Travelling Salesman Problem

**Header.** The input of the TSP Problem consists of the coordinates of each node, and the architecture uses a single fully connected layer to map these 2 features to the desired embedding dimension.

**Decoder.** The input for the TSP Problem consists of two components: (1) Last Node Encoding: The encoded representation of the last node, described by embedding dim features. (2) Mask for Invalid Nodes: The mask that handles invalid nodes. The Decoder Architecture uses two fully connected linear layers: one processes the encoded representation of the first query, while the other processes the encoded representation of the last node. After passing through the linear layers, the query vectors are reshaped to prepare them for the multi-head attention mechanism. The query vector for the last node is then combined with the saved first query vector, forming the final query vector q. This combined query vector is used in the attention mechanism to consider both the first and last nodes when selecting the optimal path.

## C.2 Vehicle Routing Problem

**Header.** The input for the CVRP Problem consists of three components: (1) Depot Coordinates: The location of the depot, described by 2 features. (2) Node Coordinates: The location of each node, described by 2 features. (3) Node Demand: The demand at each node, described by 1 feature. The Header Architecture has a fully connected linear layer that processes the depot's 2-dimensional coordinates to generate its embedding. A separate fully connected linear layer embeds the 3-dimensional features of each node, which include the coordinates and demand. These embeddings are then concatenated to combine spatial and demand information, allowing the model to jointly consider the locations, delivery requirements, and capacity constraints.

**Decoder.** The input for the CVRP Problem consists of three components: (1) Last Node Encoding: The encoded representation of the last node, described by embedding dimension features. (2) Load Information: The load at each node, described by 1 feature. (3) Mask for Invalid Nodes: The mask that handles invalid nodes. The Decoder Architecture uses a fully connected linear layer which processes the concatenated input of the last node encoding and the load information. After passing through the linear layer, the query vector for the last node is reshaped using to prepare it for the multi-head attention mechanism. The saved first query vector, is not used in this case, and the final query vector q is solely formed from the query of the last node. This query vector is then used in the attention mechanism to consider the relationships between the last node and the subsequent path decisions in the context of the vehicle's load and capacity constraints.

## C.3 Split Delivery VRP

SDVRP extends CVRP by allowing split deliveries, the model structure and initial embedding steps remain consistent.

## C.4 Orienteering Problem

**Header.** The input for the OP Problem consists of three components: (1) Depot Coordinates: The location of the depot, described by 2 features. (2) Node Coordinates: The location of each node, described by 2 features. (3) Prize: The reward associated with each node, described by 1 feature. The Header Architecture has a fully connected linear layer that processes the depot's 2-dimensional coordinates to generate its embedding. A separate fully connected linear layer embeds the 3-dimensional features of each node, which include the coordinates and prize information. The node features, consisting of coordinates and prize, are concatenated and passed through the embedding layers. Finally, the processed depot and node embeddings are concatenated together to form the complete input representation for downstream tasks.

**Decoder.** The input for the OP Problem consists of two components: (1) Last Node Encoding: The encoded representation of the last node, described by embedding features. (2) Remaining Distance Information: The remaining distance to each node, described by 1 feature. (3) Mask for Invalid Nodes: The mask that handles invalid nodes. The Decoder Architecture uses a fully connected linear layer which processes the concatenated input of the last node encoding and the remaining distance information. After passing through the linear layer, the query vector for the last node is reshaped to

prepare it for the multi-head attention mechanism. Since no first query vector is used for this problem, the final query vector q is solely formed from the query of the last node. This query vector is then used in the attention mechanism to determine the optimal path considering the remaining distances to the nodes.

### C.5    PRIZE COLLECTING TSP

**Header.** The input for the PCTSP Problem consists of two components:(1)Depot Coordinates: Represents the starting point, characterized by 2 features.(2)Node Coordinates: Represent cities in the problem, each described by 4 features, including attributes such as prizes (rewards for visiting the node) and penalties (costs for not visiting it).The Header Architecture has a fully connected linear layer processes the depots 2-dimensional features to generate its embedding and a separate fully connected linear layer embeds the 4-dimensional features of each node.The embedded node features are concatenated with their corresponding prize and penalty values, forming an augmented feature tensor.Finally, the processed depot and node embeddings are merged to construct the complete input representation for downstream computation.

**Decoder.** The input for the PCTSP Problem consists of two components: (1) Last Node Encoding: The encoded representation of the last node, described by embedding features. (2) Reward Constraint: The additional reward constraint information associated with each node, described by 1 feature. (3) Mask for Invalid Nodes: The mask that handles invalid nodes. The Decoder Architecture uses a fully connected linear layer which processes the concatenated input of the last node encoding and the reward constraint information. After passing through the linear layer, the query vector for the last node is reshaped to prepare it for the multi-head attention mechanism. The final query vector q is solely formed from the query of the last node. This query vector is then used in the attention mechanism to determine the optimal path, considering both the last node's position and the associated reward constraint.

### C.6    STOCHASTIC PRIZE COLLECTING TSP

The architecture for SPCTSP is the same as that for PCTSP, with the difference being that SPCTSP allows for probabilistic node visits. Despite this, the model structure and initial embedding steps, remain consistent between the two.

## D    EXPERIMENTAL SETTINGS

### D.1    PROBLEM RELATIONSHIP AND GENERALIZATION MEASURE

In this study, the model is trained over 500 epochs, processing 100,000 instances per epoch across **two** tasks, with a batch size of 512. Optimization is performed using the Adam algorithm, with a learning rate of $1 \times 10^{-4}$ for model parameters, and the SGD algorithm with a learning rate of $1 \times 10^{-5}$ for rotation matrices. A key aspect of this study is task-related learning, where the model is trained on both tasks simultaneously. The relationship between the tasks is captured by calculating the cosine similarity of the gradients with respect to the model parameters. This measure reflects how aligned the tasks are in terms of their contributions to learning, which is critical for understanding how the model generalizes and adapts to multiple tasks concurrently.The model is trained on a single Nvidia H100 GPU. This setup offers valuable insights into the problem's relationships and the model's generalization capabilities, with an emphasis on its adaptability and robustness throughout different training phases.

### D.2    IN-DISTRIBUTION SOLVING PERFORMANCE

In this study, the model is trained over 500 epochs, processing 100,000 instances per epoch across **six** tasks, with a batch size of 512. Optimization is carried out using the Adam algorithm with a learning rate of $1 \times 10^{-4}$ for model parameters and the SGD algorithm with a learning rate of $1 \times 10^{-5}$ for rotation matrices. The key focus of this study is on In-distribution Solving Performance, where the model is trained using collaborative multi-task learning across six tasks. During training, the model learns to simultaneously handle all six tasks. Afterward, the encoder layer parameters are fine-tuned

on one of these tasks using the Adam algorithm with a learning rate of $1 \times 10^{-5}$. Initially, the encoder layers are frozen for 100-150 epochs, and then all model parameters are fine-tuned together on the chosen task for another 100-150 epochs. This setup allows for evaluating the model's performance in solving a single task after multi-task learning. The model is trained and evaluated on a single Nvidia H100 GPU, and this setup provides valuable insights into the models in-distribution solving capabilities, focusing on its adaptability and robustness in solving seen-tasks within the distribution it was trained on.

## D.3   CROSS-PROBLEM GENERALIZATION

In this study, the model is trained over 500 epochs, processing 100,000 instances per epoch across **five** tasks, with a batch size of 512. Optimization is carried out using the Adam algorithm with a learning rate of $1 \times 10^{-4}$ for model parameters and the SGD algorithm with a learning rate of $1 \times 10^{-5}$ for rotation matrices. The key focus of this study is on In-distribution Solving Performance, where the model is trained using collaborative multi-task learning across five tasks. During training, the model learns to simultaneously handle all five tasks. Afterward, the encoder layer parameters are fine-tuned on a sixth, unseen task which is using the Adam algorithm with a learning rate of $1 \times 10^{-5}$. Initially, the encoder layers are frozen for 100-150 epochs, and then all model parameters are fine-tuned together on the chosen task for another 100-150 epochs. This setup allows for evaluating the model's ability to generalize to a task outside the distribution of the five training tasks. The model is trained and evaluated on a single Nvidia H100 GPU, and this setup provides valuable insights into the models cross-problem generalization capabilities, focusing on its ability to adapt and perform on an unseen problem after training on multiple related tasks.

Table 1: Results on TSPLIB 50–150. Models are trained on other COPs and fine-tuned on TSP-50.

|          | AM       | MCOMAB   | GCNCO   |
|----------|----------|----------|---------|
| eil51    | 15.216%  | 1.723%   | **0.973%** |
| berlin52 | 4.237%   | 1.134%   | **0.547%** |
| st70     | 1.601%   | **0.913%** | 1.752%  |
| eil76    | 2.123%   | 2.628%   | **1.654%** |
| pr76     | 0.764%   | 1.145%   | **0.816%** |
| rat99    | **2.297%** | 2.954%  | 2.451%  |
| kroA100  | 4.002%   | 3.127%   | **2.291%** |
| rd100    | 3.121%   | 2.674%   | **2.118%** |
| eil101   | 2.754%   | 2.216%   | **1.763%** |
| lin105   | **1.616%** | 3.647%  | 1.922%  |
| pr107    | 3.884%   | 3.195%   | **1.971%** |
| bier127  | 6.349%   | 7.521%   | **4.782%** |
| ch130    | 2.668%   | 2.145%   | **1.973%** |
| pr144    | 7.429%   | 5.913%   | **4.007%** |
| kroA150  | 3.612%   | **2.067%** | 3.324%  |
| average  | 4.112%   | 2.866%   | **1.935%** |

## D.4   ADDITIONAL EXPERIMENT ON TSPLIB

To evaluate the models ability to transfer from synthetic data to real benchmark instances, we conduct an additional study on the TSPLIB corpus. Concretely, all models are first trained on five COPs drawn from our main suite and then fine-tuned only on TSP-50 before being testedwithout further adjustmenton TSPLIB instances containing between 50 and 150 nodes. This protocol isolates the encoders capacity to generalize across both distribution and size shifts.

**Results.** Table 1 indicates that GCNCO attains the lowest optimality gaps on nearly every TSPLIB instance and delivers the best overall average, whereas MCOMAB(Wang et al., 2025) performs second and AM(Kool et al., 2018) lags behind. These outcomes confirm that the gradient-consistent backbone learned on diverse COPs transfers robustly to classical benchmarks with varying structures and scales.

