# OpenReview forum: "One Direction to Rule Them All: Toward Generalizable Solving Strategies Across Combinatorial Optimization Problems"
_ICLR.cc/2026/Conference — ICLR 2026 Conference Withdrawn Submission_

### Official Review · Reviewer_gaqK · 2025-10-20

**Soundness:** 3
**Presentation:** 3
**Contribution:** 2
**Rating:** 2
**Confidence:** 5

**Summary:**

This paper explores the possibility of learning common strategies for solving multiple routing combinatorial optimization problems. The main idea is to use a shared encoder with problem-specific headers and decoders. The paper also investigates the potential for transferring the pretrained encoder to new, unseen tasks.

**Strengths:**

1. The idea of using gradient homogenization appears to be original in the domain of NCO.

2. The multi-task model is trained with RL, the training process does not require labeled data.

**Weaknesses:**

1. Lack of novelty. This approach with task-specific headers and decoders, and a common encoder is presented in [1]. Furthermore, the same idea has been well-studied in [2], even in a more general way (more CO problems, richer experiments, and transfer to new problems) in [1]. The method presented in this work seems to be essentially the same, with only slightly different terminology. What the authors of this paper refer to as an "encoder", the authors in [1] called a "backbone"; the "headers" and "decoders" in this paper are equivalent to the "input and output adapters".  The proposed gradient homogenization methods are also well-known and have been investigated in the domain of multi-task training.

2. This paper claims to propose generalizable solving strategies across combinatorial optimization problems, but the experiments are conducted only on six routing problems that share many common features, and it is not clear whether this approach would work beyond the routing domain. Furthermore, experiments involving generalization to only one unseen task are not sufficient to support the claims about generalization.

3. Experiments on problems of size 20 and 50 are below the current standards of the NCO community, where state-of-the-art (SOTA) models typically solve problems with hundreds or even thousands of nodes. The baselines are very modest, and many existing multi-task models for routing are missing from the list of baselines (e.g., [3], [4], [5], together with [2]).

[1] Wang et al. Efficient Training of Multi-task Neural Solver for Combinatorial Optimization, TMLR 03/2025

[2] Drakulic et al. GOAL: A Generalist Combinatorial Optimization Agent Learner, ICLR 2025

[3] Berto et al. RouteFinder: Towards Foundation Models for Vehicle Routing Problems, TMLR 09/2025

[4] Li et al. CaDA: Cross-Problem Routing Solver with Constraint-Aware Dual-Attention, ICML 2025

[5] Zhou et al. MVMoE: Multi-Task Vehicle Routing Solver with Mixture-of-Experts, ICML 2024

**Questions:**

In addition to the questions raised in the weaknesses section, I would like to ask for further clarification on the following topics:

1. The overview of the model is not clear. Figure 1 illustrates that each problem has its own header and decoder, but Figure 2 again shows mappings involving the header, encoder, and decoder. In addition, the claim that different headers may process the same instance (with slight variations, such as additional constraints) is confusing. Does this mean that different parts (or features) of a single instance can pass through the headers of different problem encoders?

2. If I understand correctly, for each of the six problems, you pretrained the model on the remaining five and then tested its generalization - is that correct? In that scenario, I am not sure how zero-shot generalization is possible without any additional problem-specific adaptation. While it is clear how the pretrained encoder can be reused for solving a new, unseen task, how are the parameters of the problem-specific header and decoder initialized? During zero-shot generalization, if there is no training of these modules, it seems highly unlikely that the model would perform well with some random initialization.

3. The experimental settings are unclear. Does Table 4 represent a pretrained model on six tasks with additional fine-tuning on each individual task? It seems that you describe this around lines 407-408. What are the fine-tuning settings; for example, how many steps and how much data were used for fine-tuning? Did you apply the same fine-tuning procedure to MCOMAB? I would also like to see the model’s performance after multi-task training, without problem-specific fine-tuning.

4. The same unclear aspects apply to fine-tuning for cross-problem generalization. There is no mention of fine-tuning steps; the only claim is that “the model adapts well to new tasks when a small amount of task-specific data is available.” In Appendix D.3, you mention that fine-tuning is performed "by freezing the encoder layers for 100-150 epochs, followed by fine-tuning all model parameters on the chosen task for another 100-150 epochs". Considering that pretraining takes 500 steps, fine-tuning for 300 epochs is extremely long, looks more like training from scratch than fine-tuning. It would be useful to see how fine-tuning of your pretrained model performs versus the single-task scenario, trained from scratch for the same number of epochs.

5. You explicitly defined the TSP as an edge-decision problem - but later, at line 196, you defined the TSP solution as a permutation of nodes? So, do you really make decisions based on edges or nodes? What about other problems - are they node-decision or edge-decision problems?

6. I do not see a value in including the AM model as a baseline. First, it is not a multi-task model, and that work is seven years old. Moreover, AM has not been a state-of-the-art model for a long time.

---

### Official Review · Reviewer_aeat · 2025-10-31

**Soundness:** 2
**Presentation:** 2
**Contribution:** 2
**Rating:** 2
**Confidence:** 5

**Summary:**

This work aims to develop a foundation model capable of solving multiple vehicle routing problems (VRPs) simultaneously. The proposed approach employs a shared encoder combined with problem-specific headers and decoders across six tasks: the Travelling Salesman Problem (TSP), Vehicle Routing Problem (VRP), Split Delivery VRP (SDVRP), Orienteering Problem (OP), Prize Collecting TSP (PCTSP), and Stochastic PCTSP (SPCTSP). To enhance cross-task consistency, the method enforces gradient alignment across different problems using learnable rotation matrices. The experiments evaluate both in-distribution performance and cross-problem generalization, including zero-shot transfer and fine-tuning settings.

**Strengths:**

1. The motivation and methodology are clear.
2. The idea of a multi-task NCO solver is good.
3. The analysis of the gradient similarity is interesting.

**Weaknesses:**

1. Missing several baselines mentioned in the related work section.
2. No results are provided for 100-node synthetic instances.
3. Training details for the frozen and fine-tuning settings in Figure 5 are not described.
4. Inference time comparisons should be included in Figure 4.
5. Minor: Figures 4–6 would be clearer if presented as tables.

**Questions:**

What is the problem size in the experiment of Section 5.5?

---

### Official Review · Reviewer_V1or · 2025-11-01

**Soundness:** 3
**Presentation:** 3
**Contribution:** 2
**Rating:** 4
**Confidence:** 3

**Summary:**

This paper proposes GCNCO (Gradient-Consistent Neural Combinatorial Optimization), a multi-task framework designed to learn generalized solving strategies across diverse combinatorial optimization problems (COPs).

The key idea is to enforce gradient consistency—that the encoder’s gradient directions and magnitudes remain aligned across tasks—so that updates from one problem benefit others.
The model adopts a header–encoder–decoder architecture, with light problem-specific modules (headers/decoders) and a heavy shared encoder.
To homogenize gradients, GCNCO introduces:
- Gradient magnitude normalization (Eq. 6–8)
- Gradient direction alignment via rotation matrices (Eq. 9)

Trained jointly on six routing-type COPs (TSP, VRP, SDVRP, OP, PCTSP, SPCTSP), the model demonstrates promising zero-shot and fine-tuning generalization to unseen tasks.

**Strengths:**

1) Clear architectural design
  - The separation between problem-specific and problem-agnostic components (header/encoder/decoder) is clean and aligns with modular multi-task learning practice.

2) Solid empirical scope within routing COPs
  - Six representative COPs (deterministic and stochastic) are covered, demonstrating consistent performance improvements over baselines.

3) Gradient analysis as an interpretability tool
  - The use of cosine similarity heatmaps between encoder gradients gives qualitative insight into inter-problem relationships—an appealing analysis direction rarely seen in NCO papers.

**Weaknesses:**

1) Limited differentiation from prior work.
  - The approach is conceptually close to Rotograd (Javaloy & Valera 2021), extended to the header–encoder–decoder setting. Compared to MCOMAB (Wang et al., 2025), the main novelty is the alignment mechanism, but the discussion of why gradient alignment outperforms task selection is superficial.

2) Restricted experimental diversity.
  - All six tasks are routing variants; other combinatorial families (e.g., knapsack, scheduling, matching) are missing, which weakens the claim of “generalizable solving strategies across COPs.”

**Questions:**

1) Can you provide quantitative evidence that higher gradient cosine similarity indeed correlates with better zero-shot performance across problems?

2) How many fine-tuning iterations or epochs were required for the model to reach its reported post-adaptation performance? and Does gradient homogenization during pretraining empirically accelerate convergence during fine-tuning, or only improve final performance?

3) How does GCNCO perform on non-routing COPs (e.g., knapsack, scheduling?

---

### Official Review · Reviewer_NcGk · 2025-11-03

**Soundness:** 3
**Presentation:** 3
**Contribution:** 3
**Rating:** 6
**Confidence:** 3

**Summary:**

This work presents GCNCO, a framework designed to overcome the single-task limitation of most neural solvers for combinatorial optimization problems (COPs). The central idea is to train a shared, powerful encoder on multiple tasks simultaneously, using a novel technique called "gradient consistency." By optimizing special rotation matrices and loss weights, the framework forces the learning signals from different COPs to align, ensuring that the encoder develops a unified and transferable problem-solving strategy. Experiments across six routing problems show that this approach is highly effective. A key finding is that the pre-trained GCNCO encoder can be applied to a new, unseen problem and achieve performance comparable to a specialist model trained from scratch, highlighting its potential as a foundational tool for the field.

**Strengths:**

1. To break the isolation of single-task solvers, this paper introduces GCNCO, which learns a general optimization strategy using a shared encoder trained with an innovative gradient consistency mechanism.

2.Validated across six COPs, the model shows superior performance and strong generalization.   It not only outperforms baselines on the problems it was trained on but also shows exceptional cross-problem generalization.   Its effectiveness is validated by comprehensive experiments: gradient similarity analysis confirms the alignment works, while in-distribution tests show no performance degradation on trained tasks

3. Its pre-trained encoder achieves competitive zero-shot results on unseen tasks and surpasses baselines after fine-tuning, demonstrating its potential as a foundational model for rapid, resource-efficient optimization.

**Weaknesses:**

1. Unproven Scalability to Larger Instances:
The experiments are confined to small-scale problems (20-50 nodes), so the model's performance on larger, industrially relevant instances remains an open question.

2. The title is "ONE DIRECTION TO RULE THEM ALL: TOWARD GENERALIZABLE SOLVING STRATEGIES ACROSS COMBINATORIAL OPTIMIZATION PROBLEMS". However, currently all problems are routing based. A clarification on this could be helpful.

**Questions:**

1.Task Selection and Similarity: The six COPs chosen for the experiments (TSP, VRP, SDVRP, OP, PCTSP, SPCTSP) are all variants of routing problems on graphs. How does the choice of included tasks affect
learning? If a new problem is very different (say, a scheduling problem), would the
encoder still transfer? Have you tried adding a completely unrelated COP to the
training set, and if so, how does it impact the shared encoder?

2. Scalability to Larger Problem Instances:
The experiments are conducted on relatively small problem sizes (20 and 50 nodes). While common for foundational research in neural combinatorial optimization, the performance on these scales does not guarantee effectiveness on larger, more industrially relevant instances (e.g., hundreds or thousands of nodes). The computational complexity of the attention-based encoder and the overhead of the bi-level optimization might become prohibitive as the problem size increases.

3. A discussion on robustness of the model when used different training size data of problems are used could be helpful as in reality different sized problems occur commonly.

---

### Note · Authors · 2025-12-17

I have read and agree with the venue's withdrawal policy on behalf of myself and my co-authors.